# A stochastic analysis of the interplay between antibiotic dose, mode of action, and bacterial competition in the evolution of antibiotic resistance

**Peter Czuppon**[1,2,3]*, **Troy Day**[4], **Florence Débarre**[2], **François Blanquart**[3]

**1** Institute for Evolution and Biodiversity, University of Münster, Münster, Germany, **2** Institute of Ecology and Environmental Sciences of Paris, Sorbonne Université, UPEC, CNRS, IRD, INRA, Paris, France, **3** Center for Interdisciplinary Research in Biology, CNRS, Collège de France, PSL Research University, Paris, France, **4** Department of Mathematics and Statistics, Department of Biology, Queen's University, Kingston, Canada

* p.czuppon@uni-muenster.de

**Data Availability Statement:** Code and data to reproduce the figures, have been deposited at https://github.com/pczuppon/AMRWithinHost.

## Abstract

The use of an antibiotic may lead to the emergence and spread of bacterial strains resistant to this antibiotic. Experimental and theoretical studies have investigated the drug dose that minimizes the risk of resistance evolution over the course of treatment of an individual, showing that the optimal dose will either be the highest or the lowest drug concentration possible to administer; however, no analytical results exist that help decide between these two extremes. To address this gap, we develop a stochastic mathematical model of bacterial dynamics under antibiotic treatment. We explore various scenarios of density regulation (bacterial density affects cell birth or death rates), and antibiotic modes of action (biostatic or biocidal). We derive analytical results for the survival probability of the resistant subpopulation until the end of treatment, the size of the resistant subpopulation at the end of treatment, the carriage time of the resistant subpopulation until it is replaced by a sensitive one after treatment, and we verify these results with stochastic simulations. We find that the scenario of density regulation and the drug mode of action are important determinants of the survival of a resistant subpopulation. Resistant cells survive best when bacterial competition reduces cell birth and under biocidal antibiotics. Compared to an analogous deterministic model, the population size reached by the resistant type is larger and carriage time is slightly reduced by stochastic loss of resistant cells. Moreover, we obtain an analytical prediction of the antibiotic concentration that maximizes the survival of resistant cells, which may help to decide which drug dosage (not) to administer. Our results are amenable to experimental tests and help link the within and between host scales in epidemiological models.

## Author summary

Antibiotic treatment creates a beneficial environment for the evolution of antibiotic-resistant bacterial strains. The dosage of the antibiotic drug during treatment plays an

**Funding:** P.C. has received funding from the European Union's Horizon 2020 research and innovation program under the Marie Sklodowska-Curie grant agreement PolyPath 844369. F.B. is funded by the ERC StG 949208 EvoComBac. F.D. is funded by ANR-19-CE45-0009-01 TheoGeneDrive. The funders had no role in study design, data collection and analysis, decision to publish, or preparation of the manuscript.

**Competing interests:** The authors have declared that no competing interests exist.

important role during this process. Here, we derive analytical predictions for the survival probability of a resistant subpopulation until the end of treatment with either a biostatic, i.e. growth-inhibiting, or a biocidal, i.e. death-promoting, drug. Importantly, we obtain a prediction for the antibiotic concentration that maximizes this survival probability. Additionally, we also compute the size of the resistant subpopulation at the end of treatment and its carriage time after treatment until it gets outcompeted by an antibiotic-sensitive strain. This post-treatment phase is relevant only for commensal bacteria. We find that treatment with a biocidal drug, compared to a biostatic drug, increases the risk of resistance evolution, results in a larger resistant subpopulation size at the end of treatment and prolongs the carriage time, and therefore shedding, of the resistant strain. Our analytical predictions can be tested experimentally and link the within-host and the population scale of antibiotic resistance dynamics.

## Introduction

Bacterial pathogens resistant to antibiotics are a major public health challenge responsible for more than a million deaths per year [1]. Ecological and evolutionary principles have guided the design and evaluation of strategies to reduce the use of antibiotics while still eradicating pathogenic bacteria in patients [2]. Theoretical and experimental research on antibiotic resistance has focused on the optimal drug dose, e.g. [3–5] and the optimal prescription regimen in hospitals, e.g. [6–9] to limit the evolution and spread of resistance (reviewed in [10–12]).

The question of the optimal drug dose and duration to limit the evolution of resistance has received much attention. "Hitting early and hitting hard" [13, 14] may limit the emergence of drug resistance [15–18]. More recent studies challenge this view [2, 3, 19] based on the result that the probability of emergence of drug resistance is maximized at an intermediate concentration. A low antibiotic concentration prevents the emergence of resistance by allowing the maintenance of the sensitive strain, which impedes growth of the resistant strain through competition. On the other hand, a high antibiotic concentration also prevents the emergence of resistance by quickly eradicating the bacterial population, limiting the input of resistance mutations, and directly limiting the growth of the resistant subpopulation. Thus, the emergence of drug resistance is most likely maximal at an intermediate concentration where the sensitive population is eradicated, which frees resistance from competition ("competitive release", [20]), allowing the resistant subpopulation to grow. Depending on the treatment window, which describes the feasible range of drug concentration and is defined by clearance of the infection-causing bacteria and considerations on drug toxicity, a low or a high drug dose may be best to limit the emergence of resistance. This line of argument applies to symptomatic infections caused by pathogenic bacteria, which are the direct target of antibiotic treatment. It also applies to commensals and opportunistic pathogens that are carried asymptomatically, which are not the target of the antibiotic treatment but under 'bystander selection' [21].

The survival or extinction of a small resistant subpopulation emerging during treatment is ultimately governed by the random processes of bacterial cell division and death. It is not clear how this stochasticity impacts the early dynamics of resistance. Within the vast field of pharmacokinetics/pharmacodynamics, the field studying the dynamics of antibiotic concentration and the impact of antibiotics on bacterial cells, several studies modeled the dynamics of sensitive and resistant cells *in vitro* or *in vivo* [22–25]. These models describe deterministically how a resistant subpopulation of cells can grow and cause a rebound in bacterial population size. The deterministic description of the bacterial dynamics is justified when a substantial resistant sub-population

exists prior to treatment, as may be the case with large mutation rates from sensitivity to resistance and/or large initial population sizes. However, in many common situations in commensalism or in infection, the resistant bacterial population is initiated from one or a small number of resistant cells, and mutations or gene transfers conferring resistance rarely occur. The early phase of resistance emergence, when the resistant subpopulation is still small, is a stochastic process. Yet only a handful of studies theoretically explored the impact of stochasticity on the emergence of drug resistance [3, 4, 11]. The simple realization that resistance emergence is a stochastic process has recently inspired elegant empirical and theoretical work measuring and computing the probability of emergence in single cell assays [26]. Yet, an analytical solution for the probability of emergence of resistance during treatment is still missing (although numerical solutions for resistance survival probabilities under periodic antimicrobial treatment conditions were obtained recently [27, 28]). Such solution would be important to characterize the drug concentration maximizing the probability of emergence of resistance, the size of the resistant subpopulation within the host and, in the case of commensal bacteria, how long a treated host carries and sheds resistance after treatment. These results can in turn provide insight on the optimal drug dose and inform between-host models describing the shedding and transmission of the resistant strain to other hosts. A stochastic description of the evolution of drug resistance also requires that we specify how bacterial cells compete, and whether the drug impedes cell division or actively kills cells (biostatic or biocidal). Both are important determinants of the probability of emergence of resistance. Little attention has previously been paid to these aspects of the life cycle of bacterial cells.

Here, we analyze a stochastic model of within-host dynamics and treatment, and derive new analytical results on the survival probability of a resistant subpopulation until the end of treatment; secondarily, we analyze the size of the resistant subpopulation at the end of treatment, and the carriage time after treatment.

## Model

We study the population dynamics of a bacterial population in the absence and presence of an antibiotic drug. We first describe how a drug initially clears the sensitive population, potentially allowing the establishment and rise of a resistant subpopulation. This first phase applies both to the bystander antibiotic exposure of bacteria that are carried asymptomatically and to the direct treatment of infections (symptomatic disease). A single resistant cell is introduced during treatment by mutation or transmission from another host. This resistant subpopulation may or may not establish in the host. If it does establish, it grows logistically until the end of treatment. Following this establishment phase, we model the dynamics leading to the extinction of the resistant subpopulation after treatment. When bacteria are carried asymptomatically, no further treatment is prescribed after the end of the antibiotic course. Sensitive strains can re-establish, either because they were not fully eradicated or because they are reintroduced by transmission. The sensitive strain eventually competitively excludes the resistant subpopulation because of the cost of resistance. This potentially requires several re-introductions of the sensitive strain. When bacteria instead caused an infection, it is reasonable to assume that a second course of antibiotic will be applied if the bacterial load is still high at the end of the first course. We do not model this scenario.

Overall, the model is a stochastic description of the transient establishment, peak and extinction of a resistant subpopulation within the treated host (Fig 1).

### Bacterial population dynamics

We study two different *models* of density regulation of the bacterial population (Table 1). In the first model, population density affects the birth rate, which we refer to as 'birth

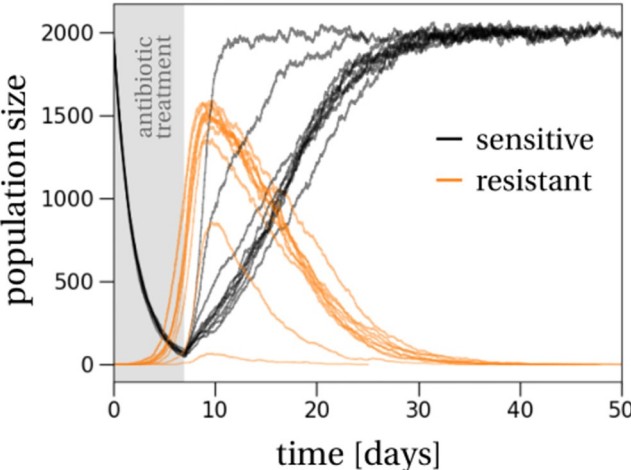

**Fig 1. Population dynamics of the resistant (orange) and sensitive (black) strains.** The sensitive subpopulation declines almost deterministically during treatment, which is administered for seven days (gray shaded region). At the same time the resistant subpopulation, if it survives, increases in size. There is strong variation between the resistant growth curves because of stochastic birth and death events. Non-surviving trajectories are omitted. After treatment, both subpopulations grow until the overall pathogen population reaches the carrying capacity, which happens approximately two days after the end of treatment when the resistant subpopulation reaches its maximum. The overall size of the resistant subpopulation at that point shows variation among the ten sample trajectories, again because of demographic stochasticity. This variation carries over to the time at which the resistant subpopulation is outcompeted by the sensitive subpopulation in the absence of antibiotic treatment.

competition'. For example, limited resource availability impedes cell division [29]. In the second model, population density increases the death rate, which we refer to as 'death competition'. For example, bacterial cells may secrete antibiotics, toxins or viruses that kill neighbouring cells [30–33]. Additionally, death competition could implicitly model the effect of the immune system [3, 4]. Both models of density regulation yield the same deterministic population dynamics, as long as the overall population size remains below the carrying capacity. In the stochastic model formulation and analysis, however, the exact form of density regulation matters as we will show below. The relative contributions of the two forms of competition in natural environments is poorly known. It is certain that nutrients are limited and therefore birth competition occurs; however, the extent to which bacteria kill each other when at high density in the host environment is to the best of our knowledge unknown.

Additionally, we distinguish between two *modes of action* of antibiotics: biostatic and biocidal. A biostatic drug, e.g. tetracycline or erythromycin, reduces the birth (division) rate of the cells. A biocidal drug, e.g. ciprofloxacin or streptomycin, increases the death rate. The deterministic dynamics are the same for the two modes of action if the antibiotic

**Table 1. Birth and death rates in the four studied scenarios.** The overall birth and death rates, denoted by $\lambda_j$ and $\mu_j$, are composed of the birth, death and competition processes that occur at rates $\beta_j$, $\delta_j$ and $\gamma_j$, respectively. Additionally, the effect of antibiotics, denoted by $\alpha_j(c)$, affects either the birth or the death rate, depending on the type of drug administered. The variable $c$ denotes the concentration of the antibiotic and the index $j$ indicates the strain-specificity, $j$ = $S$ or $j$ = $R$ for antibiotic-sensitive or -resistant cells.

| | | birth competition | death competition |
|---|---|---|---|
| **biostatic** | $\lambda_j(x_S, x_R)$ | $\max(\beta_j - \gamma_j(x_S + x_R) - \alpha_j(c), 0)$ | $\max(\beta_j - \alpha_j(c), 0)$ |
| | $\mu_j(x_S, x_R)$ | $\delta_j$ | $\delta_j + \gamma_j(x_S + x_R)$ |
| **biocidal** | $\lambda_j(x_S, x_R)$ | $\max(\beta_j - \gamma_j(x_S + x_R), 0)$ | $\beta_j$ |
| | $\mu_j(x_S, x_R)$ | $\delta_j + \alpha_j(c)$ | $\delta_j + \gamma_j(x_S + x_R) + \alpha_j(c)$ |

concentration is small enough. They differ, however, for larger concentrations because the birth rate cannot be smaller than zero, while the death rate can, in principle, increase without bounds as the antibiotic concentration increases (Fig A in S1 Appendix). Interestingly, the stochastic dynamics always depend on the mode of action because the variance of the underlying stochastic process is different for the two modes of action.

The two density regulation models and two modes of action define four *scenarios*. We start by describing the dynamics of the sensitive strain in the bacterial population. The per capita birth and death rate of sensitive cells are denoted by $\lambda_S(x_S, x_R)$ and $\mu_S(x_S, x_R)$, where $x_S$ and $x_R$ denote the densities of the sensitive and resistant subpopulations, respectively. These rates defining the stochastic process vary across scenarios. For simplicity, we mostly present results from the main scenario: birth competition and biocidal treatment. In this main scenario the birth and death rates are given by:

$$\lambda_S(x_S, x_R) = \max(\beta_S - \gamma_S(x_S + x_R), 0) \quad \text{and} \quad \mu_S(x_S, x_R) = \delta_S + \alpha_S(c), \tag{1}$$

where $\beta_S$ denotes the birth rate of the sensitive strain, $\delta_S$ the intrinsic death rate of the sensitive strain, $\gamma_S$ the competition parameter of the sensitive strain and $\alpha_S(c)$ the effect of the antibiotic treatment on the sensitive strain, where the drug is administered at concentration $c$ (more details in the next section). The basic death rate comprises both cell death and the outflux of the host compartment colonized by bacteria.

The dynamics of the resistant strain are defined analogously. We assume that resistance comes at a cost [34] that is mediated through a reduced birth rate $\beta_R < \beta_S$, an increased death rate $\delta_R > \delta_S$, or through less competitiveness $\gamma_R > \gamma_S$. Being (partly) resistant to the antibiotic reduces the rate at which the resistant type is affected by the antibiotic, i.e. $\alpha_R(c) < \alpha_S(c)$.

The description of the four scenarios at the individual cell level is summarized in Table 1. In Section H in S1 Appendix we additionally study another model of density regulation, where the antibiotic interacts with the competition process described by the density-dependent terms, and in Section J in S1 Appendix we study an alternative model with an explicit host immune response instead of death competition between the two bacterial subpopulations.

## Antibiotic response curve

We assume that the antibiotic affects the population dynamics of both the sensitive and the resistant strains. The *Minimum Inhibitory Concentration* (MIC) is the concentration at which the net growth rate, measured during the exponential phase (when density regulation can be neglected) is zero. For the sensitive strain, it is denoted by $\text{mic}_S$ and given by: $\beta_S - \delta_S - \alpha_S(\text{mic}_S) = 0$. The MIC of the resistant strain, $\text{mic}_R$, is defined analogously and is larger than the MIC of the sensitive strain. The resistant strain is unaffected by treatment (fully resistant), when $\text{mic}_R$ equals infinity.

The antibiotic response curve, denoted $\alpha_j(c)$ ($j$ either $S$ or $R$), defines the effect of the antibiotic as a function of its concentration. In line with empirical studies [35–38], we assume a sigmoid function relating the antibiotic-mediated death rate to the antibiotic concentration:

$$\alpha_j(c) = (\psi_{j,\max} - \psi_{j,\min}) \frac{\left(\dfrac{c}{\text{mic}_j}\right)^{\kappa}}{\left(\dfrac{c}{\text{mic}_j}\right)^{\kappa} - \dfrac{\psi_{j,\min}}{\psi_{j,\max}}} . \tag{2}$$

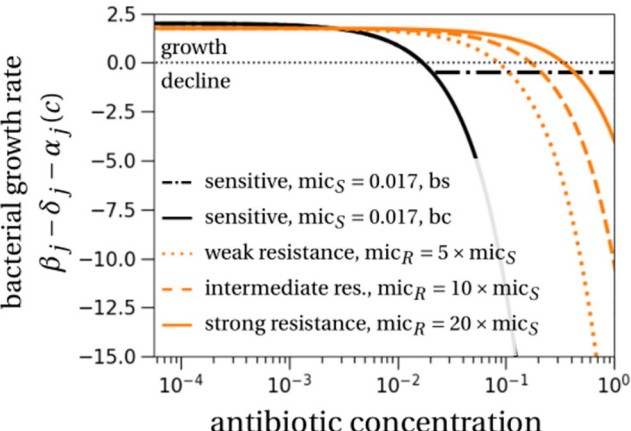

**Fig 2. Bacterial growth rate without density-dependent effects for different antibiotic concentrations.** The curves show the effect of a biostatic (bs) and biocidal (bc) drug. For a biostatic drug, we plot $\max(\beta_j - \alpha_j(c), 0) - \delta_j$ (instead of the expression written on the y-axis label, $\beta_j - \delta_j - \alpha_j(c)$). For clarity, we only show the biostatic drug effect for the *sensitive* strain (black dot-dashed line), and not for the resistant strains. The effect of a biostatic drug on the resistant strains would yield a minimal growth rate at $-\delta_R$. Parameters for the sensitive antibiotic response curve are motivated by estimates for ciprofloxacin from [35]: $\beta_S = 2.5$ per day ($d^{-1}$), $\beta_R = 2.25$ ($d^{-1}$), $\delta_S = \delta_R = 0.5$ ($d^{-1}$), $\kappa = 1.1$, $\psi_{j,\min} = -156 \times \log(10)$ ($d^{-1}$), $\psi_{j,\max} = \beta_j - \delta_j$ ($d^{-1}$).

The function increases from zero when the antibiotic concentration is zero ($c = 0$), to $\psi_{j,\max}$ when $c = \mathrm{mic}_j$, and to $\psi_{j,\max} - \psi_{j,\min}$ (note that $\psi_{j,\min} < 0$) when the antibiotic concentration is much larger than $\mathrm{mic}_j$.

The parameter $\psi_{j,\max} = \beta_j - \delta_j$ is the maximal growth rate of strain $j$. This parameter is can-celled by antibiotic-induced death when the concentration is at the MIC. The parameter $\psi_{j,\min}$ is negative and corresponds to the maximal decline rate of strain $j$, i.e., the value $\beta_j - \delta_j - \alpha_j(\infty)$. The parameter $\kappa$ is the steepness of the antibiotic response curve. Fig 2 shows the resulting pathogen growth rate under antibiotic treatment, i.e. $\beta_j - \delta_j - \alpha_j(c)$, for our default parameter values. For simplicity, we neglect pharmacokinetics and model a constant antibiotic concentration during treatment.

## Parameterization

Although we emphasize that our analytical results allow general insights, we explored in simu-lations a set of parameters that is biologically plausible for an infection by Enterobacterales. As explained above, the key parameters governing the response to antibiotics ($\mathrm{mic}_S$, $\kappa$, $\psi_{j,\min}$) are informed by the empirical studies of *E. coli* dynamics under ciprofloxacin treatment [35] (note that $\psi_{j,\max}$ is not a free parameter but equal to the maximal growth rate in the absence of anti-biotic treatment). We test different values for the level of resistance (MIC value) of the resistant strain. As has been shown before, evolution can increase the resistance levels by several orders of magnitude, at least up to 1000-fold for certain bacteria [39]. We did not change the shape of the antibiotic response curve for the resistant strain, in line with experimental findings [40]. Regarding demographic parameters, the birth rate of the sensitive strain is $\beta_S = 2.5$ $d^{-1}$. This corresponds to a doubling time of 6–7 hours, or around four generations per day, similar to what was measured for *Salmonella enterica* acute systemic infection in a murine model [41]: 2.4 $d^{-1}$. We assumed that the cost of resistance decreases the birth rate of resistant strains to $\beta_R = 2.25$ $d^{-1}$. Fitness is experimentally measured by competitive assays with 24 hours of growth, which for example in the case of *E. coli* includes 5–6 hours of exponential growth on average. Defining fitness as the relative exponential growth difference in six hours ($\exp(\beta_R \times 0.25)/\exp$

($\beta_S \times 0.25$)), our choice of parameters corresponds to a $\sim 6\%$ fitness disadvantage of the resistant strain, which is in line with experimental data [42]. The death rates in the absence of antibiotic effect are set to $\delta_S = \delta_R = 0.5\ d^{-1}$, which approximately corresponds to the bacterial death rates estimated in liver and spleen in [41]: liver = 0.41 $d^{-1}$, spleen = 0.22 $d^{-1}$.

In addition, in Section G of the SI we study a second parameter set reflecting the lifestyle of *E. coli* in commensalism in the gut. This parameter set assumes a faster doubling time of $\sim 90$ minutes ($\beta_S = 11\ d^{-1}$) [43]. The death rates are set to the same values as for infection, $\delta_S = \delta_R = 0.5\ d^{-1}$. Here, this reflects plausible values for the outflux of *E. coli* in the human gut corresponding to a mean transit time of 2 days [44].

The parameter determining the strength of competition between bacteria is set to $\gamma_S = \gamma_R = 1\ d^{-1}$. A larger competition parameter increases the negative effect of the wild type on the mutant and thus reduces the survival probability, the size of the resistant subpopulation at the end of treatment and the carriage time. Lastly, the order of the relevant population size is set to $K = 1,000$ for computational feasibility. The value of the population size $K$, which can be understood as the volume that the population inhabits, does not influence the survival probability of resistance (our main result), and influences the final resistant population size in a straightforward way as a multiplicative factor. However, it affects the mean carriage time of the resistant strain after the end of treatment in a non-obvious way, by modulating the magnitude of the stochastic fluctuations.

## Stochastic simulations

In the stochastic simulations, we keep track of bacterial counts, denoted by $X_j$ ($j$ either $S$ or $R$) and not densities, $x_j$, as introduced above. To translate between counts and densities, we divide the count by the order of the population size $K$. The relationship between densities and counts is then given by $x_j = X_j/K$.

The population updates in the stochastic simulations are determined by the exact Gillespie algorithm [45]. To this end, the birth and death rates of sensitive and resistant cells are computed based on the current population size by the formulas in Table 1. Random numbers to determine the next update, birth or death of a sensitive or resistant cell, and the time of the next update are drawn from a uniform and exponential distribution, respectively, and the population is updated. More details are provided in Section K in S1 Appendix.

All simulations are written in the C++ programming language and use the GNU Scientific Library. Code and data to reproduce the figures, which have been generated with python, have been deposited at https://github.com/pczuppon/AMRWithinHost.

## Results

We compute the probability of survival of a resistant subpopulation until the end of treatment, the size of the resistant subpopulation at the end of treatment and the carriage time of the resistant strain after the end of treatment. Together, these quantities provide a comprehensive picture of the stochastic population dynamics of a resistant subpopulation within a host (Fig 1).

### Survival probability of the resistant strain during treatment

We compute the survival probability of a resistant strain emerging during treatment. We assume that the resistant subpopulation establishes from a single resistant cell appearing during treatment. This single resistant cell could arise from a *de novo* mutation, or equivalently from transmission from another host during treatment, or from standing genetic variation. For clarity, in the main text we just present the results of resistance evolution from standing

genetic variation, that is, when one resistant cell is present at the beginning of treatment. Results on resistance dynamics from *de novo* emergence are discussed in Section F in S1 Appendix. Qualitatively, the results of standing genetic variation and *de novo* emergence of resistance during treatment are similar. One exception is that biostatic drugs applied at concentrations above the MIC of the sensitive strain result in no resistance evolution because cell replication is fully suppressed, i.e., the survival probability is equal to zero for these concentrations (Figs F-H in S1 Appendix). Another difference is that with *de novo* resistance the maximal survival probability is shifted to lower concentrations compared to resistance emergence from standing genetic variation. In our main parameterization, the maximum is reached at or below the MIC of the sensitive strain in the case of *de novo* emergence.

The dynamics of the resistant subpopulation emerging from a single resistant cell are well described by a stochastic birth-death process. We note that these dynamics only depend on the *density* of sensitive cells (not their absolute numbers), which means that they are independent of the choice of the parameter *K*. For small numbers of resistant cells, a deterministic description through an ordinary differential equation is not appropriate because stochastic fluctuations cannot be ignored and extinction events cannot be observed. We use a branching process in a time-heterogeneous environment [46–48] to approximate the probability of survival until the end of treatment. This is the probability of having at least one resistant cell in the bacterial population at the end of treatment, which we refer to as *emergence* of resistance. Emergence, or survival, therefore does not imply establishment. We say that resistance *established* if the resistant type has risen to a density large enough that the probability of stochastic loss is negligible. A similar argument has been used in [3] to distinguish between emergence and establishment (note that they refer to emergence as "occurrence" and to establishment as "emergence"). The difference between emergence and establishment becomes visible when we study the size of the resistant subpopulation at the end of treatment below.

In the following, we qualitatively compare the predictions for the survival probability in the four scenarios. Theoretical predictions are derived in Section B in S1 Appendix. In general, the survival probabilities depend on the exponential growth rates of the two strains, denoted by $\rho_j = \beta_j - \delta_j - \alpha_j(c)$, and the selection coefficient $s = \rho_R - \rho_S$. For example, under birth competition and biocidal treatment we find the following form of the survival probability, denoted by $\varphi$, for an infinitely long treatment (Eq. (B.35) in S1 Appendix):

$$\varphi = \frac{s\rho_S\rho_R}{s\rho_R\rho_S + (\delta_R + \alpha_R)\rho_S(\gamma x_S(0) + s)} = \frac{1}{1 + \underbrace{\frac{\delta_R + \alpha_R}{\rho_R}}_{\text{stochastic death}} \underbrace{\left(\frac{\gamma x_S(0)}{s} + 1\right)}_{\text{competition}}} . \tag{3}$$

This concise equation is our main result and illuminates the factors influencing the resistant survival probability. It encapsulates, in a simple form, the chance that a single resistant cell survives treatment when it appears in a sensitive subpopulation of density $x_S(0)$, and the sensitive subpopulation is declining under the action of biocidal treatment. The survival probability is large when the factor in the denominator is small. This depends on two processes. The first is stochastic variability: the stochastic death of resistant cells threatens the survival of the resistant subpopulation. This is translated in mathematical terms by the contribution of death terms to the overall growth rate (the term $(\delta_R + \alpha_R)/\rho_R$). Of course, if resistant cells do not die ($\delta_R + \alpha_R = 0$), the survival probability is 1. The second process is competition, mediated through the term $\gamma x_S(0)/s$, which depends on the initial sensitive density and the strength of competition $\gamma$. It scales with a factor $1/s$, which means that competition is alleviated when the growth rate difference between resistant and sensitive cells is large ($s = \rho_R - \rho_S$). Similar formula structures

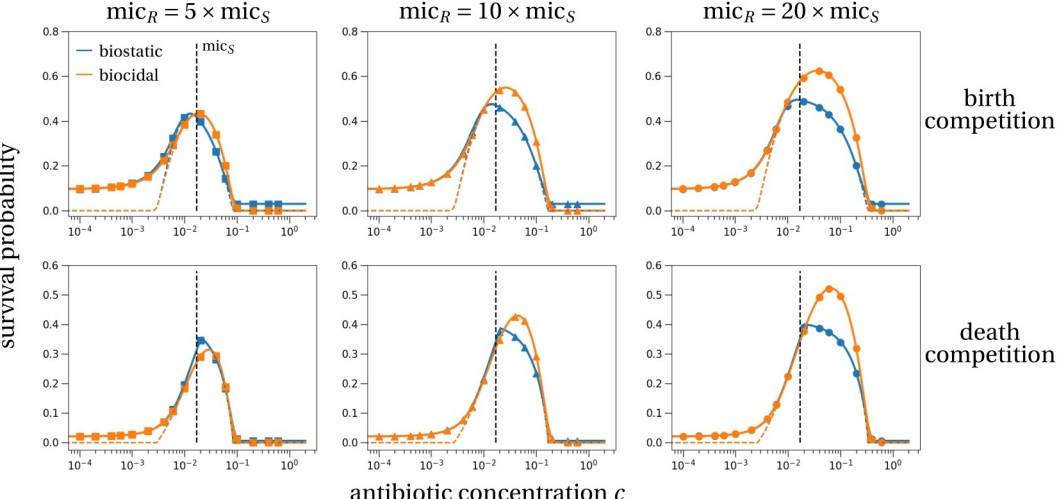

**Fig 3. Survival probability of the resistant subpopulation for varying drug concentrations, MIC values, drug types and density regulations.** At the beginning of treatment the population consists of the sensitive strain at its carrying capacity and a single resistant cell. Then treatment, either with a biostatic (blue) or biocidal (orange) antibiotic, is applied for seven days (solid lines) or infinitely long (colored dashed lines). The vertical dashed line indicates the MIC of the sensitive strain, $\text{mic}_S = 0.017$. The survival probability of the resistant subpopulation obtained from $10^6$ stochastic simulations (symbols) agrees perfectly with our theoretical predictions (Section B in S1 Appendix). The different rows show different models (Table 1): top row = birth competition; bottom row = death competition. The columns show survival probabilities for different values of the resistant MIC as a multiple of the sensitive MIC: left = low, middle = intermediate, right = high resistant MIC. Parameter values are: $\beta_S = 2.5$ per day $(d^{-1})$, $\beta_R = 2.25$ $(d^{-1})$, $\delta_S = \delta_R = 0.5$ $(d^{-1})$, $\kappa = 1.1$, $\psi_{j,\min} = -156 \times \log(10)$ $(d^{-1})$, $\psi_{j,\max} = \beta_j - \delta_j$ $(d^{-1})$, $\gamma_S = \gamma_R = 1$ $(d^{-1})$, $K = 1,000$, $X_R(0) = 1$, $X_S(0) = (\beta_S - \delta_S)K/\gamma$.

are found in both scenarios of death competition (Eqs. (B.9) and (B.13) in S1 Appendix), the scenario of birth competition and biostatic antibiotic is not possible to resolve analytically by our method (Eq. (B.31) in S1 Appendix). Analogous formulae can be obtained for a finite treatment duration $\tau$. For details, we refer to Section B in S1 Appendix.

In Fig 3, we plot the survival probabilities of the resistant subpopulation for different MICs of the resistant strain and in the different models of density regulation and antibiotic modes of action as defined in Table 1. The survival probability of the resistant subpopulation is maximal for intermediate antibiotic concentrations $c$, in line with previous results [3, 4]. The relationship quantitatively depends on the model of density dependence. The survival probability is always higher for birth rather than death competition (compare $y$-axes between the different rows in Fig 3). This is explained by a larger variance of the stochastic process when the density affects the death rate. Intuitively, a larger variance increases stochastic fluctuations, which reduces the survival probability for a population with large birth and death rates compared to a population with the same deterministic growth rate but small birth and death rates [49].

By the same argument, one would expect that biocidal antibiotic treatment results in a smaller survival probability than biostatic treatment. Interestingly though, we observe that in all scenarios, there is just a small difference between the two antibiotic modes of action (compare the blue and orange curves and symbols in Fig 3). In fact, survival probabilities tend to be larger for biocidal drugs. The largest difference is for highly resistant strains (right column in Fig 3). This is explained by the much stronger competitive release caused by biocidal drugs. For biocidal drugs, increasing the antibiotic concentration continues to increase the death rate of sensitive strains and the strength of competitive release (Fig 2, black solid line). For biostatic drugs in contrast, the sensitive population will reach its minimal growth rate for concentrations just slightly above the MIC. This deterministic effect dominates the stochastic effect

explained above and explains why the survival probability of resistant strains is larger for a biocidal treatment.

## Predicting the antibiotic concentration that maximizes the resistant survival probability

As just outlined, the survival probability exhibits a maximum at intermediate antibiotic concentrations. In the context of treatment of symptomatic infections, the exact location of this antibiotic concentration would inform whether the "hit hard" strategy is optimal to limit resistance evolution.

Despite having explicit expressions for the survival probability (Section B in S1 Appendix), we were not able to analytically find the concentration that maximizes the survival probability of the resistant subpopulation during treatment. However, under biocidal treatment we can derive implicit solutions that depend on the demographic parameters. For birth competition and biocidal treatment for example, the concentration maximizing the survival probability of the resistant subpopulation is found by solving the following equality (details in Section C in S1 Appendix):

$$\frac{\alpha'_S(c)}{\alpha'_R(c)} - 1 = \frac{\beta_R(\rho_R(c) - \rho_S(c))}{\rho_R(c)(\delta_R + \alpha_R(c))} + \frac{\beta_R(\rho_R(c) - \rho_S(c))^2}{x_S(0)\gamma\rho_R(c)(\delta_R + \alpha_R(c))} \,, \tag{4}$$

where $\alpha'_j(c)$ denotes the derivative of the antibiotic response curve with respect to the antibiotic concentration, and $\rho_j = \beta_j - \delta_j - \alpha_j(c)$ is the exponential growth rate under treatment of strain $k \in \{S, R\}$ (as in Eq (3)). A similar condition can also be derived in the scenario of a biocidal drug and death competition (Section C in S1 Appendix).

The left-hand side of Eq (4) reflects how much more the antibiotic affects the growth of the sensitive strain compared to the resistant strain. This difference can be positive or negative. For low concentrations, it will be positive as the sensitive strain is more affected by the antibiotic than the resistant strain. As soon as the antibiotic response for the resistant strain has a steeper negative slope than for the sensitive strain, the left-hand side will become negative. It is not possible to generally predict when this will be the case, but one can expect this to be at concentrations above the MIC of the sensitive strain (at least for sigmoid antibiotic response curves). The right-hand side is always positive as long as the selection coefficient (the growth rate difference $\rho_R(c) - \rho_S(c)$) is positive, which is always the case in a neighborhood of the maximizing concentration. Still, it is not straightforward to predict when these fractions are large and no general prediction of the maximizing concentration is possible.

For biostatic drugs, we are not able to derive a similar condition. For death competition, however, we find an upper bound for the concentration maximizing the survival probability. Denoting this concentration by $\tilde{c}$, it is given by the lowest concentration that ensures no further sensitive cell replication (details in Section C in S1 Appendix):

$$\tilde{c} = \inf\{c : \beta_S - \alpha_S(c) = 0\} \,. \tag{5}$$

Importantly, this implies that this upper bound is independent of the level of resistance of the resistant strain. The maximum of the survival probability remains at the same concentration, as can be verified visually in Fig 3 (lower row, blue curves). Note that if the intrinsic death rate of sensitive cells, $\delta_S$, is zero, we find $\tilde{c} = mic_S$, which implies that the maximum of the resistant survival probability is at a concentration equal to or less than the MIC of the sensitive type (note that in Fig 3 we have $\delta_S > 0$). Intuitively, this upper bound is the concentration at which competition pressure with sensitive cells is maximally reduced. Any increase in antibiotic concentration will result in the same decline of the sensitive strain. Administering

concentrations beyond this critical threshold therefore only decreases the birth rate of the resistant strain, explaining the decline in survival probability above this critical concentration. In the scenario of treatment with a biostatic drug and birth competition, no analytical prediction on the survival-maximizing concentration seems possible.

## Size of resistant subpopulation at end of treatment

We now study the *size* of the resistant subpopulation when it survives treatment. The probability of survival is useful to know if any resistant subpopulation is able to emerge during treatment. Yet, the amount of shedding of resistant cells from one host to another will depend not only on the possible emergence but also on the absolute size reached by the resistant subpopulation. Note that the resistant subpopulation may still continue to grow after treatment has ended if the total carrying capacity has not been reached at that time (Fig 1).

The resistant subpopulation size at the end of treatment is typically larger in a stochastic model than in the deterministic counterpart (compare solid and dotted lines in Fig 4). Precisely, because we condition on survival of the resistant subpopulation, the stochastic trajectory will increase faster initially than the deterministic trajectory until a certain threshold number of cells is reached. This threshold is defined by the survival probability being equal to one for a resistant subpopulation of that size [50, 51]. From that threshold level on, the dynamics are equal to the deterministic system of ordinary differential equations. The detailed mathematical analysis is stated in Section D in S1 Appendix. The difference between the stochastic and deterministic predictions is largest for small differences between the resistant and sensitive strain because this is where the survival probability of the resistant strain is smallest, which speeds up the initial stochastic resistance establishment dynamics the most (compared to deterministic dynamics).

In Fig 4, we compare the stochastic prediction for the size of the resistant subpopulation at the end of treatment (solid lines) and the deterministic prediction (dotted lines) with simulation results (symbols). Under birth competition, the two modes of action of antibiotics affect

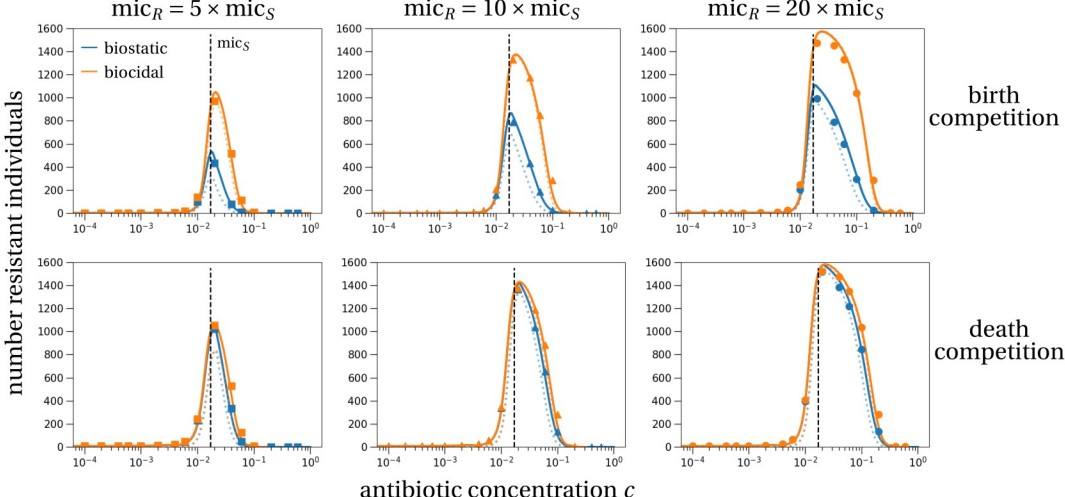

**Fig 4. Size of the resistant subpopulation at the end of treatment if the resistant subpopulation survives.** At the onset of treatment there is exactly one resistant cell in the population and treatment lasts for seven days. The top row shows the results for the model of birth competition, the lower row corresponds to death competition. Dotted lines show the deterministic prediction of the resistant subpopulation size, solid lines correspond to the stochastic prediction that incorporates a 'correction' due to conditioning on survival. Symbols are the mean resistant subpopulation sizes of $10^6$ stochastic simulations that were conditioned on survival of the resistant subpopulation. Blue and orange colors correspond to biostatic and biocidal treatment, respectively. Parameters are as in Fig 3.

the resistant subpopulation size at the end of treatment differently (upper row in Fig 4). We consistently find that the size of the resistant subpopulation at the end of treatment is larger for biocidal drugs than for biostatic ones. This is again explained by the stronger competitive release under biocidal drugs: the sensitive subpopulation declines faster under biocidal treatment than under biostatic treatment (Fig A in S1 Appendix). Under death competition, the resistant subpopulation sizes at the end of treatment do not differ substantially between the two antibiotic types for concentrations below the sensitive MIC (lower row in Fig 4). For concentrations above the sensitive MIC, again the resistant subpopulation size is smaller with biostatic than biocidal treatment. However, the difference between the two treatments is smaller than under birth competition. This is explained by the decline of the sensitive population with biostatic treatment being stronger under death competition than under birth competition, which in turn increases the competitive release effect and therefore the resistant subpopulation size.

The largest resistant subpopulation is reached for an antibiotic concentration close to the MIC of the sensitive strain in all parameter sets and scenarios (compare dashed vertical line with the maximum of the curves in Fig 4).

## Carriage time of resistant subpopulation within a host after treatment

So far, we have studied the dynamics of a bacterial population, commensal or pathogenic, during antibiotic treatment. Following treatment, the resistant subpopulation will, if it established, still be present and can therefore be transmitted to other hosts. Here, we study the carriage time of a resistant commensal strain in a host after treatment has ended. The analysis of this last phase is less relevant for infections by pathogenic bacteria, because in this case treatment would be continued, possibly with a different antibiotic, to reduce the pathogen load and to cure the patient.

The carriage time of a commensal resistant subpopulation can be interpreted as the time it takes a sensitive strain to re-establish within a host after antibiotic treatment has ended. At the end of treatment, the antibiotic concentration is set to $c = 0$ and therefore $\alpha_S(0) = \alpha_R(0) = 0$. In the absence of antibiotics, the resistant subpopulation will likely be replaced by the sensitive strain because the resistant strain has a fitness cost compared to the sensitive strain. The size of the resistant and sensitive subpopulations at the end of treatment (previous section) are the initial conditions for this post-treatment phase. If the sensitive subpopulation has been eradicated during treatment, we assume that it restarts with a single cell that is introduced from the host environment immediately after treatment has ended. In reality, the time until re-introduction of sensitive cells is a stochastic process governed by the influx rate from the environment.

We estimate the extinction time of the resistant subpopulation, conditioned on its extinction. To characterize this phase, we apply a timescale separation of the fast ecological and slow evolutionary dynamics, which reduces the problem to a single dimension [52, 53]. We follow the frequency dynamics of the resistant type in the population, while the overall population size is assumed to remain (approximately) constant. A timescale separation of the population size and the frequency dynamics applies if the difference between the resistant and sensitive strains is small or even negligible compared to the ecological rates [54, 55]. In our standard parameter set, the evolutionary rate is proportional to the cost of resistance $\beta_S - \beta_R$. This cost is 10% of the ecological rate $\beta_k - \delta_k$, which translates to 6% in terms of the empirical fitness cost in [42], as outlined in the Parameterization section above. We first outline the deterministic population dynamics and then comment on the differences with a fully stochastic version. The deterministic population dynamics unfold as follows (Fig 1): first, on the fast timescale, the

 Evolution of antibiotic resistance under drug mode of action and bacterial competition

overall population size, which is the sum of sensitive and resistant cells, will rebound to the carrying capacity very quickly. Once at carrying capacity, the population will slowly move towards the extinction boundary of the resistant strain due to its fitness disadvantage. In other words, the population size remains approximately constant while the frequency of the sensitive strain increases. With demographic fluctuations, which are due to stochasticity in the birth and death processes, extinction of the resistant strain is not certain as the sensitive strain can go extinct on its way to re-establishment. Still, we assume that eventually the sensitive strain will replace the resistant subpopulation. Biologically, this is motivated by a constant influx of sensitive cells, so that one of these repeated establishment attempts will eventually be successful. We therefore condition the stochastic process on extinction of the resistant subpopulation. Because of this conditioning, the carriage will in reality be at least as long as our estimate. Applying results from one-dimensional stochastic diffusion theory [50, 56, 57], we compute the mean extinction time of the resistant population (mathematical details are stated in Section E in S1 Appendix).

The comparison between our theoretical prediction and the simulation results in Fig 5 shows that the timescale separation in our competitive Lotka-Volterra model captures well the simulation results, even for this relatively large evolutionary rate compared to the ecological processes. At low antibiotic concentrations, the carriage time is comparable for the two types of antibiotics. At concentrations above the sensitive MIC (vertical dashed line in Fig 5), the carriage time is larger after biocidal treatment than after biostatic treatment. This is explained by a larger relative frequency of the resistant strain in the bacterial population at the end of treatment (Fig C in S1 Appendix). This difference arises again because the sensitive population size decreases faster with biocidal treatment than with biostatic treatment (Fig A in S1 Appendix). This increases the frequency of the resistant strain directly by the lower sensitive population size and indirectly through a larger competitive release effect, which may result in a higher resistant population size at the end of treatment.

Stochastic effects do not reduce much the carriage time compared to a deterministic analysis (Fig D in S1 Appendix). The carriage time of resistant strains is substantially reduced by

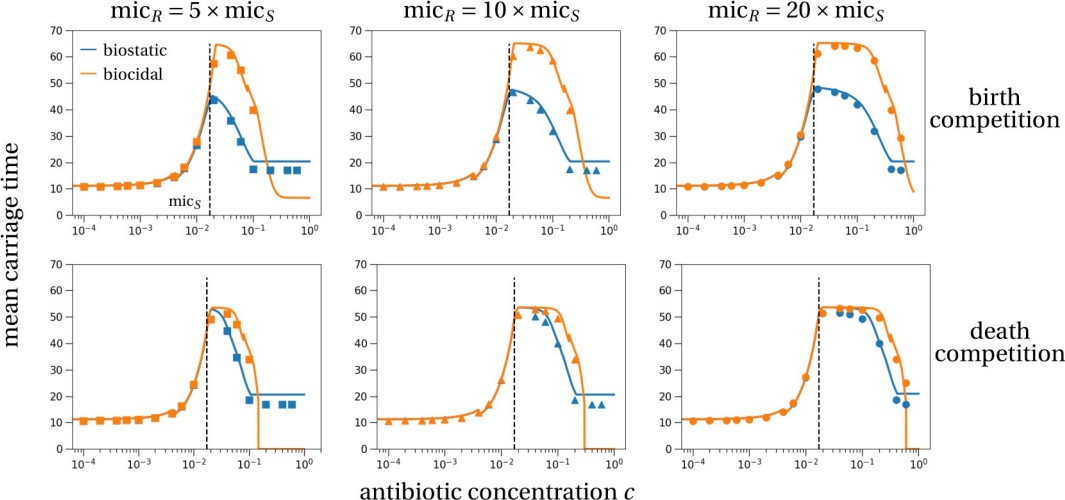

**Fig 5. Mean carriage time of the resistant subpopulation in a host (including the treatment time $\tau$ = 7).** The carriage time is set to zero if the resistant subpopulation did not survive antibiotic treatment. Discontinuities are a result of discontinuities in the estimate of the resistant subpopulation size at the end of treatment (discussion in Section D in S1 Appendix). The theoretical predictions (lines) are derived in Section E. Symbols show the average carriage time of $10^6$ stochastic simulations. The color coding and figure structure is the same as in the previous figures. Parameters are the same as in Fig 3.

stochasticity only when the variance of the stochastic process is large. This is the case under death competition or when the cost of resistance is smaller than in the default parameter set (Fig E in S1 Appendix).

## Discussion

We developed a fully stochastic model describing the dynamics of antibiotic resistance in a treated host. We mainly analyzed the survival probability of the resistant strain until the end of treatment. We also derived results on the resistant subpopulation size at the end of treatment. These results apply to commensal bacteria as well as to pathogens. Additionally, we computed the time after treatment until a commensal is replaced by an antibiotic-sensitive strain due to competition, which we refer to as the carriage time.

Generally, the survival probability, size of resistant subpopulation, and carriage time are all maximized at an intermediate antibiotic concentration (Figs 3–5) in line with previous studies [3, 4]. Our main new results are explicit analytical formulae for the probability of survival and predictions for the antibiotic concentration that maximizes this probability. We find that in the scenario of biostatic treatment and death competition, this maximizing concentration is very close to, not necessarily below, the MIC of the sensitive strain, and independent of the resistant pharmacodynamic parameters. The distance between the sensitive MIC and the maximizing concentration is determined by the death rate of the sensitive strain and the shape of the antibiotic response curve of the sensitive strain. Precisely, the smaller the death rate of the sensitive strain, the closer is this maximizing concentration to the sensitive MIC. Under biocidal treatment, the condition for the concentration maximizing resistant survival cannot be evaluated in all generality and depends in particular on the shape of the antibiotic response curve (Eq (4)). We find in extensive simulations that the concentration maximizing the resistant survival probability is always close to the sensitive MIC. This was true in our default and alternative parameter set (Section G in S1 Appendix), for a broad range of sigmoid antibiotic response curves (Section I in S1 Appendix), in an alternative model where the bacterial population is limited by the host's immune response instead of intraspecific competition (Section J in S1 Appendix) and in previous models, e.g. [3, 4, 11]. If patients are generally prescribed antibiotic doses at or above the MIC of the sensitive drug, this implies that to limit resistance emergence, "hitting hard" (at the maximum tolerable dose) is best in the case of biostatic drugs and seems often to be best in the case of biocidal drugs. More importantly, hitting hard is best not only to limit the probability of emergence of resistance, but also to limit the growth of the resistant subpopulation (Fig 4).

One originality of our model is that we describe in detail the bacterial life cycle including density regulation on birth vs. death, and antibiotic effect on birth vs. death (see [27] for similar models). These considerations are particularly important in stochastic models. One interesting result is that the probability of survival is larger when competition reduces the birth rate. This is explained by a reduction in demographic stochasticity from this type of density regulation compared to death competition. Precisely, in two scenarios with the same deterministic dynamics, as is the case with birth and death competition for a growing resistant subpopulation, in the scenario with the lower birth and death rates, in our case birth competition, the subpopulation is less likely to be lost due to demographic stochasticity [49]. Another interesting result is that biocidal drugs lead to stronger competitive release than biostatic drugs. Biocidal drugs eliminate the sensitive strain faster by driving the death rate to very large values, while the birth rate, which is affected by biostatic drugs, can only be reduced to zero (Fig 2 and Fig A in S1 Appendix). As a consequence, the probability of emergence of resistance, resistant subpopulation size and overall carriage time of resistant pathogens are larger for biocidal drugs.

This result that biostatic drugs suppress resistance evolution more than biocidal drugs, derived when resistant cells are already present at the onset of treatment (Fig 3), is reinforced when resistance instead evolves *de novo* (Section F in S1 Appendix). As biostatic drugs suppress cell replication, they also limit the mutational input compared to biocidal drugs. These comparisons between biostatic and biocidal drugs corroborate and extend findings from another theoretical study that investigated *de novo* resistance evolution under periodic treatment with different drug types [27]. The authors found that infection clearance is more likely under biocidal treatment. Yet, perfect biostatic drugs, i.e. drugs that fully suppress cellular division, are superior in suppression of *de novo* resistance evolution. These conflicting theoretical recommendations suggest that it might be complicated to translate our results to the clinical setting. The simple and robust result emerging from theory on the superior impact of biocidal drugs in faster clearance of the sensitive strain is not even verified in recent clinical studies. Several meta-analyses found no difference in treatment success between biostatic and biocidal drugs [58–60]. It is possible that drugs that are in theory biostatic, in practice also directly kill bacteria at clinically relevant doses [59]. *In vitro* experiments might be a promising next step to test the conflicting impacts of different modes of action on clearance and resistance evolution (see below).

In our parameterization, we assumed high drug resistance (large MIC differences between resistant and sensitive cells), which leads to large survival probabilities (up to 60%) and limited stochastic effects. We also assumed a large, but realistic, cost of resistance ($\sim 6 - 10\%$ in main text and S1 Appendix) [42], which reduced stochastic effects in the post-treatment phase. All stochastic effects would be stronger for smaller differences between the drug-sensitive and -resistant cells, i.e., weak resistance and weak cost of resistance. Precisely, the survival probability would be smaller and the differences between different antibiotic modes of action and density dependence would be larger. A smaller cost of resistance will always prolong the carriage time (Fig E in S1 Appendix).

Our model has several limitations. We model a single antibiotic course of fixed duration (e.g. seven days). Variation in treatment duration will affect the quantitative values, but not the qualitative pattern of the studied quantities. For example, compare the dashed and solid colored lines in Fig 3 that correspond to an infinite treatment and a seven day treatment, respectively. Importantly, the concentration maximizing the risk of resistance evolution remains unchanged. The last phase examining the duration of carriage of the resistant strain is less relevant if we consider pathogenic bacteria. In this case, as the total bacterial population size quickly recovers to pre-treatment values after the end of treatment (Fig 1), it is likely that the host would undergo a second antibiotic course to cure the disease. However, bystander exposure to antibiotics in carriage (not infection), where our post-treatment phase applies, is the most common context of exposure for several important bacterial species [21]. Last, we assume that the antibiotic concentration is constant throughout treatment. In reality, the antibiotic concentration might fluctuate in time, which would impact the probability of emergence and the final population size. Models with explicit pharmacokinetics did not directly study the probability of resistance emergence and establishment in comparison to a scenario with constant concentration of a single drug [28, 61–63]. It is therefore difficult to speculate how explicit pharmacokinetics would affect our results.

Our theoretical work suggests several interesting experiments. Some of our findings have in fact already been tested experimentally. A larger resistant subpopulation size at the beginning of treatment increases the probability of survival and establishment of the resistant strain and the subpopulation size at the end of treatment ([26] and Eq. (B.2) in S1 Appendix). Another study investigated the population dynamics of a pathogen population under different drug modes of action and found that biocidal treatment reduces the population size more strongly than biostatic treatment ([64] and our Fig A in S1 Appendix). Based on our theoretical results,

further *in vitro* experiments could be conducted to characterize the probability of emergence of resistance depending on drug concentration, the model of density regulation and the drug mode of action. We predict that the differences between the two drug modes of action are strongest when density affects the birth rate (Fig 4).

The validity of our prediction on the resistant survival probability in Eq (3) can also be tested. To this end, one needs to expose sensitive cells to antibiotics, and measure the survival probability of an introduced resistant cell as a function of the antibiotic concentration. In parallel, one can evaluate all terms of Eq (3) through simple *in vitro* experiments. The exponential growth rates of both types of cells ($\rho_S$, $\rho_R$), as well as the death rate of antibiotic resistant bacteria ($\alpha_R$), can all be measured *in vitro* at different antibiotic concentrations. The competition coefficient $\gamma$ need not be measured provided that the starting sensitive population is at stationary phase equilibrium (the term $\gamma x_S(0)$ is equal to $\beta_S - \delta_S$). Our prediction that biostatic drugs always have a maximizing concentration at (or slightly above) the sensitive MIC, irrespective of the level of resistance, can even be tested without knowledge of the demographic parameters.

The within-host dynamics of antimicrobial resistance underpin the between-host epidemiological dynamics of resistance. In fact, the dynamics of colonization by resistant strains and the slow dynamics of clearance of the resistant strain after treatment are key determinants of the intermediate equilibrium frequency reached by the resistant strain in a host population [65]. Here, we produced mathematical results that help bridge the gap between the two scales. All quantities derived here (probability of survival, population size reached, carriage time) are relevant to epidemiological dynamics and determine the total shedding of resistance. Stochasticity implies that not all events of transmission of a resistant strain lead to survival and establishment of a resistant subpopulation; when establishment is successful, however, a greater resistant subpopulation is reached than in the equivalent deterministic model. These two effects may approximately compensate in terms of shedding of resistance: accounting for stochasticity leads to fewer hosts colonized by resistant strains, but to more resistance transmission per successfully colonized host. The carriage time is reduced by stochasticity, but this reduction is small for our choice of parameters (highly resistant strains with a strong cost; Figs D-E in S1 Appendix).

In conclusion, we developed a fully stochastic mathematical description of the emergence of a resistant subpopulation during antibiotic treatment. This work could motivate experiments studying how drug resistance evolves from small populations of resistant cells, and better epidemiological models explicitly linking the within and between host scales.

## Supporting information

**S1 Appendix. Mathematical analysis of the model.** The supplement contains details of the mathematical derivation of the main results and additional simulations and figures.
(PDF)

## Acknowledgments

We are grateful to the High Performance Computing service of the University of Münster for providing computational resources on the computer cluster PALMA II. We are also thankful to the INRAE MIGALE Bioinformatics Facility (MIGALE, INRAE, 2018; Migale Bioinformatics Facility, https://doi.org/10.15454/1.5572390655343293E12) for providing computational resources during the early phase of this project.

## Author Contributions

**Conceptualization:** Peter Czuppon, Florence Débarre, François Blanquart.

**Data curation:** Peter Czuppon.

**Formal analysis:** Peter Czuppon, François Blanquart.

**Funding acquisition:** Peter Czuppon, Florence Débarre, François Blanquart.

**Investigation:** Peter Czuppon, Troy Day, Florence Débarre, François Blanquart.

**Methodology:** Peter Czuppon.

**Software:** Peter Czuppon.

**Supervision:** Florence Débarre, François Blanquart.

**Validation:** Peter Czuppon, Troy Day, Florence Débarre, François Blanquart.

**Visualization:** Peter Czuppon.

**Writing – original draft:** Peter Czuppon, François Blanquart.

**Writing – review & editing:** Peter Czuppon, Troy Day, Florence Débarre, François Blanquart.

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
