## [Decision Letter · Decision Letter 0]

31 Mar 2023

Dear Dr. Czuppon,

Thank you very much for submitting your manuscript "Stochastic within-host dynamics of antibiotic resistance: Resistant survival probability, size at the end of treatment and carriage time after treatment" for consideration at PLOS Computational Biology.

As with all papers reviewed by the journal, your manuscript was reviewed by members of the editorial board and by several independent reviewers. In light of the reviews (below this email), we would like to invite the resubmission of a significantly-revised version that takes into account the reviewers' comments.

In light of the reviewer comments and from our own reading of the manuscript, it appears that the manuscript as written fails to meet the "biological signficance" threshold required by the journal. It is not clear whether the numberical and anlytical results are applicable to the real-lfe situation, whether the parameters chosen are realistic and how the the novel theoretical development are called by failures of simpler models. Therfore, a significant revision would be required for the manuscript to be appropriate for the journal.

We cannot make any decision about publication until we have seen the revised manuscript and your response to the reviewers' comments. Your revised manuscript is also likely to be sent to reviewers for further evaluation.

Sincerely,

Oleg A Igoshin

Academic Editor

PLOS Computational Biology

Amber Smith

Section Editor

PLOS Computational Biology

Reviewer's Responses to Questions

**Comments to the Authors:**

Reviewer #1: This work performed full stochastic modeling of how resistant and sensitive populations grow and die during antibiotic treatment. Authors carefully considered various processes, e.g., density regulation, modes of action, transmission, immune response, etc. The resulting mathematical model is extensive, described in detail in Sup Mat (in about 35 pages!). I think this model could be useful to the community.

However, as it is written now, it is very hard to understand the focus of the work. After reading a whole article, it is still not clear why this approach is better than previous work. Stochastic modeling could be more realistic. There are various processes (as considered here) that indeed take place during antibiotic treatment. While conventional pharmacodynamic models are inherently deterministic and could be potentially limited, they extensively use experimental data, are practical and are widely used. It is not clear why the authors' framework is better. For example, under what conditions stochastic modeling is relevant? I think that authors should clarify and emphasize the new insight that this approach was uniquely position to provide, which previous work failed to do.

There are some numerical data shown for some parameter ranges. But, without discussing why those parameters were chosen and relevant, it is hard to evaluate the importance of the findings. In fact, there are a lot of basic parameters that were taken as it is and were not tuned (e.g. Table A2). It seems that there were some biological assumptions when choosing those values (e.g., why is the birth rate of resistant strain is lower than that of sensitive strain but death rate the same?). Authors should clarify these issues up front.

Reviewer #2: The authors study the stochastic dynamics of a pathogen population comprising a sensitive and a resistant strain that is subjected to

antibiotic treatment. The focus is on the probability of survival of the resistant strain until the end of the treatment, as well

as on the time required for the sensitive strain to subsequently replace the resistant strain. The main text describes selected

results for a few typical scenarios. An extensive supplement contains mathematical derivations and explores the robustness of the

main results with regard to modifications of the modeling assumptions. Overall, this is a thorough and well-written study of

an important problem that is principally suited for publication in PLoS Computational Biology. Nevertheless, I believe that a

revision addressing the points listed below would considerably increase the impact and usefulness of the work.

1. To what extent can the four scenarios in Table 1 be mapped to actual pathogen-drug combinations? While I assume that the

distinction between biostatic and biocidal drugs can be made on the basis of the mechanism of drug action, is it known under

what conditions the competition between bacteria affects primarily the birth or the death of cells?

2. Related to this, the way in which the population dynamics are implemented in the stochastic simulations needs to be specified.

Currently no information about this is provided, except for the link to the code (line 146). This is not sufficient. A more

comprehensive description of the stochastic simulations is particularly important for those scenarios in the supplement that are only

explored computationally (e.g., Section F).

3. What is the motivation for the choice of carrying capacity/population size (K=1000)? To me, a bacterial population of 1000

cells would seem to be rather small. Nevertheless, even for this choice the difference between stochastic and deterministic

treatments is not very pronounced in most cases. Does this mean that for larger (and possibly more realistic) population

sizes the deterministic theory would often be sufficient?

4. Lines 171 ff: The authors define "emergence" of resistance as the survival of the resistant strain until the end of the treatment

and emphasize that this is not identical to the notion of "establishment" used, e.g., in Ref.22. While this is true, I wonder if

in practice the two aren't essentially equivalent, in the sense that survival without prior establishment would seem to be

very unlikely (it would require the resistant subpopulation to remain at small population numbers through the treatment

phase). Perhaps this question could be addressed by simulations?

5. Throughout the manuscript and the supplement, the MIC of the susceptible strain is shown in the concentration plots by a

vertical dotted line. Although a corresponding label can usually be found in the first panel of each figure, I missed this

information when first reading the manuscript. Please mention this explicitly either in the text or in the caption

of the first figure where it is used.

Reviewer #3: The review is uploaded as an attachment.

**Have the authors made all data and (if applicable) computational code underlying the findings in their manuscript fully available?**

Reviewer #1: Yes

Reviewer #2: Yes

Reviewer #3: Yes

PLOS authors have the option to publish the peer review history of their article (what does this mean?). If published, this will include your full peer review and any attached files.

Reviewer #1: No

Reviewer #2: No

Reviewer #3: No
---

## [Decision Letter · Decision Letter 1]

17 Jul 2023

Dear Dr. Czuppon,

We are pleased to inform you that your manuscript 'A stochastic analysis of the interplay between antibiotic dose, mode of action, and bacterial competition in the evolution of antibiotic resistance' has been provisionally accepted for publication in PLOS Computational Biology. Thanks for taking the revieer comments seriously and doing a thorough job on revision! (OAI)

Best regards,

Oleg A Igoshin

Academic Editor

PLOS Computational Biology

Amber Smith

Section Editor

PLOS Computational Biology

Reviewer's Responses to Questions

**Comments to the Authors:**

Reviewer #1: I am happy with the revision.

Reviewer #2: The authors have addressed all issues raised in my previous report in a satisfactory manner. I recommend publication of the manuscript in its present form.

Reviewer #3: The authors have addressed my comments and modified the manuscript accordingly, for which I thank them. The quality of the manuscript has improved considerably, and I am sure this work will interest many colleagues.

**Have the authors made all data and (if applicable) computational code underlying the findings in their manuscript fully available?**

Reviewer #1: None

Reviewer #2: Yes

Reviewer #3: Yes

PLOS authors have the option to publish the peer review history of their article (what does this mean?). If published, this will include your full peer review and any attached files.

Reviewer #1: **Yes: **Minsu Kim

Reviewer #2: **Yes: **Joachim Krug

Reviewer #3: No

---

## [Editor Report · Acceptance letter]

9 Aug 2023

PCOMPBIOL-D-23-00292R1 

A stochastic analysis of the interplay between antibiotic dose, mode of action, and bacterial competition in the evolution of antibiotic resistance

Dear Dr Czuppon,

I am pleased to inform you that your manuscript has been formally accepted for publication in PLOS Computational Biology. Your manuscript is now with our production department and you will be notified of the publication date in due course.

With kind regards,

Zsofi Zombor
